# Transient Positive SARS-CoV-2 PCR without Induction of Systemic Immune Responses

**DOI:** 10.3390/vaccines11020482

**Published:** 2023-02-19

**Authors:** Barbara C. Gärtner, Verena Klemis, Tina Schmidt, Martina Sester, Tim Meyer

**Affiliations:** 1Department of Medical Microbiology and Hygiene, Saarland University, Building 43, Kirrberger Straße, D-66421 Homburg/Saar, Germany; 2Department of Transplant and Infection Immunology, Saarland University, Building 47/77, Kirrberger Straße, D-66421 Homburg/Saar, Germany; 3Institute of Sports and Preventive Medicine, Saarland University, Campus, Geb. B 8.2, D-66123 Saarbrücken, Germany

**Keywords:** immune response, mucosal, T cell immunity, colonization, COVID-19, football

## Abstract

SARS-CoV-2 testing is dominated by PCR to guide treatment and individual as well as public health preventive measures. Among 1700 football (soccer) players and staff of the German Bundesliga and Bundesliga 2 who were regularly tested by PCR twice weekly, 98 individuals had a positive PCR (May 2020 to mid-January 2021). A subset of these were retested shortly after the initial positive result. Among those, 11 subjects were identified who only had a transient single positive PCR of low viral load. All individuals were asymptomatic and none developed long COVID. We tested SARS-CoV-2 IgG and IgA as well as SARS-CoV-2 specific CD4 und CD8 positive T cells, and showed that only one out of 11 individuals developed SARS-CoV-2 specific cellular and humoral immunity after the positive PCR, whereas a specific immunity was undetectable in all other individuals. Thus, a single positive PCR might indicate that transient colonization of the upper respiratory tract with SARS-CoV-2 may occur without systemic induction of specific adaptive immunity. Together with test artifacts as another potential reason for a transiently positive test, this finding may favor cautious interpretation of positive PCR results or retesting before initiating intervening treatment or infection control measures in some cases.

## 1. Introduction

Detection of SARS-CoV-2 RNA using nucleic acid amplification assays such as PCR is the cornerstone diagnostics for acute infection. At the individual level, a positive PCR is interpreted as evidence for infection and contagiousness. Moreover, in patients with risk factors antiviral treatment might be initiated to avoid severe disease and hospitalization [1]. During the first years of the SARS-CoV-2 pandemics a single positive test was generally followed by isolation and contact tracing and quarantine to prevent further spread.

When developing public health recommendations, the number of positive PCR tests can trigger specific measures such as physical distancing, limiting movements or minimizing gatherings of individuals and large groups [2,3,4]. In addition, in public health and in international restrictions such as the EU certificate [5], vaccinated individuals were equated with individuals recovered from SARS-CoV-2 infection for a certain period of time based on the suspected development of a protective immune response. Nevertheless, the status of a person recovered from infection was primarily based on a positive PCR and not on an immune response, underlining the impact of a positive PCR test result. One must keep in mind that all measures were rather strict and had a profound and sometimes negative impact on physical and mental health [6].

The local and the systemic immunity is a result of complex mechanisms involving the innate and the adaptive immune system and consists of humoral as well as cellular factors. Whereas systemic responses are usually measured in blood, local reactions are much more difficult to determine. Local immunity is a barrier that blocks further spread of infection which may not necessarily result in induction of a systemic immune response. The innate local immunity against RNA viruses is usually stimulated by the presence of viral RNA which is recognized by various pattern recognition receptors. This is followed by a cascade of events including secretion of cytokines or chemokines followed by further attraction of other cells including neutrophils, monocytes and natural killer (NK) cells. Only if innate immunity itself is not sufficient in blocking the infection, the virus will establish infection in the cell, and adaptive immune responses are induced. The adaptive immune response is dominated by antigen-specific antibodies and antigen-specific T cells, which are detectable in circulation but also play a role at the site of infection [7].

In the setting of a cohort of elite athletes and staff with twice weekly PCR testing for approximately nine months (from May 2020 until SARS-CoV-2 vaccination was introduced in mid-January 2021) we found a relevant number of individuals with a single positive PCR test that was followed by a negative test in a rather short interval questioning the interpretation of a positive PCR result as evidence for a relevant infection. As alternative explanation, this may result from false positive PCRs. In addition, local transient infections may occur, regulated by the local innate immune response at mucosal surfaces. As both alternative explanations would not necessarily be followed by induction of systemic immune responses, these individuals were further characterized for development of SARS-CoV-2 specific antibodies and T cells to provide indirect evidence for a relevant infection.

## 2. Study Group, Materials and Methods

The study participants with a transiently positive PCR originated from a cohort of elite football players and staff, where a special hygiene program was established in May 2020 during the first lockdown in Germany to facilitate the restart of the German Bundesliga and Bundesliga 2 with matches behind closed doors [8,9]. This cohort comprised approximately 1700 players and staff (mostly males (>95%), median age of approximately 30 years as described before [8]) with direct contact to players (e.g., coaches, physiotherapists) of the 36 professional soccer teams in Germany. The concept included twice weekly PCR testing during the entire football seasons starting in May 2020 using nasopharyngeal and/or oropharyngeal swabs. Swabs were taken by the team physicians or trained staff. PCRs were performed in selected laboratories accredited according to the highest quality standard for medical laboratories (DIN ISO EN 15189) and contracted by the Deutsche Fußball Liga (DFL). The labs used only high-quality commercial PCR assays (Cobas 6800/8800 SARS-CoV-2 (Roche, Basel, Switzerland), GeneExpert SARS-CoV-2 (Cepheid, Sunnyvale, CA, USA), TaqPath COVID-19 (Thermo Fisher Scientific, Waltham, MA, USA), Allplex SARS-CoV-2 (Seegene, Seoul, Republic of Korea), ampliCube SARS-CoV-2 on Lightcycler 480 II (Mikrogen, Neuried, Germany), and the quality of the assays were verified by the Coronavirus Task Force of the DFL. At least a dual target PCR with an extraction/inhibition control was mandatory. The analytical cut-offs provided by the manufacturers were used. Interpretation criteria were harmonized and tests were considered positive when at least two target genes tested positive.

In a subset of 11 individuals (all males, aged 30.6 ± 9.1 (mean ± standard deviation) years), a total of 4.7 mL of heparinized blood was drawn. The interval between positive PCR and blood sample was chosen to be at least 21 days to allow sufficient time for development of humoral and cellular immune responses. For SARS-CoV-2 specific IgG and IgA a semi-quantitative ELISA according to the manufacturer’s instructions (Euroimmun, Lübeck, Germany) was used. Antibody levels were expressed as ratios of the extinction of the sample divided by the extinction of a calibrator serum. Ratios < 0.8 were scored negative, ratios between ≥0.8 and <1.1 were scored equivocal, and ratios ≥ 1.1 were scored positive [10].

SARS-CoV-2 specific CD4 and CD8 T cells were quantified by intracellular cytokine staining using multiparameter flow-cytometry directly from whole blood after specific stimulation with overlapping peptide pools encompassing the SARS-CoV-2 parental spike protein using an in-house assay as previously described [10,11,12,13]. After 2 h of stimulation, brefeldin A was added to accumulate cytokines intracellularly. Four hours later, blood cells were processed for flow-cytometric staining using fluorescently-labeled antibodies toward CD4, CD8, CD69 and interferon-γ (IFNγ). Spike-specific T cells were quantified as CD69-IFNγ double positive cells among CD4 and CD8 T cells using a gating strategy as described before [10,11,12,13]. Stimulation with the peptide-diluent (DMSO) was used as negative control, stimulation with *Staphylococcus aureus* enterotoxin B (SEB) served as internal positive control for general T cell responsiveness as described before [10,11,12,13]. Percentages below 0.05% were scored negative.

The study was approved by the ethics committee of the Ärztekammer des Saarlandes (reference 76/20), and all individuals gave written informed consent.

## 3. Results

Approximately 1700 players and staff were regularly screened at least twice weekly by SARS-CoV-2 PCR between May 2020 and mid of January 2021. Testing was mandatory, and players with positive PCRs were excluded from active participation in professional football. During this interval, 98 individuals tested positive (players and staff). The first PCR test was found on 13 September 2020 and the last test included in this study was found on 14 January 2021. Subtyping of the variants was not done, but during the initial period, the parental Wuhan and during the last weeks of the observation the Alpha variants dominated in Germany. A positive PCR had relevant consequences for the players and the teams, as the players were excluded from training and from the next matches. To avoid these consequences, some teams closely repeated PCR testing in positive players if they considered the results to be untrustworthy although this was neither mandatory nor encouraged in the protocol of the Bundesliga [9].

Among these 98 positive individuals, 11 (all males, aged 30.6 ± 9.1 (mean ± standard deviation) years) were identified with an unexpected pattern of a single positive PCR preceded and immediately followed by a series of negative PCRs. The individuals (8 players, two coaches and one physiotherapist) originated from nine different clubs throughout Germany with only two individuals from the same two club. We wondered if this unusual pattern reflected an actual SARS-CoV-2 infection that resulted in the systemic induction of specific humoral and cellular immune response. Thus, symptoms and blood samples of these 11 individuals were collected and analyzed (Figure 1).

All subjects were asymptomatic and none developed long COVID over a follow-up of at least 2 years. Viral load of the initial positive PCR test was rather low (median Ct 36.3 (range 37.9–28.0)). A median of 42 (range 21–135) days after the positive PCR, SARS-CoV-2 specific antibodies (IgG and IgA) as well as SARS-CoV-2 specific CD4 and CD8 T cells were determined. All individuals showed reactive T cells after stimulation with the positive control stimulus SEB (CD4 median 2.43% (IQR 1.63–3.62%), CD8 median 6.82% (IQR 4.10–12.94%)) demonstrating the principal responsiveness of cellular immunity [13]. Interestingly, a systemic humoral and cellular immune response was only found in one out of 11 individuals, whereas all other tested individuals did not mount any specific immune response. Two individuals had serum samples tested for antibodies prior to the first positive PCR that were negative since they took part in another study [8].

## 4. Discussion

In this study, we report the results of a unique cohort of individuals that were monitored closely for symptoms and with PCR twice weekly over a long period of time, which is rarely done. We show that a positive PCR may occur in the absence of clinical symptoms and without development of systemic humoral or cellular immune response. Of note, the study was performed prior to vaccination. Therefore, immunological analyses were not influenced by vaccination responses. As another strength of our study, a subset of the initially positive individuals was retested with PCR in a short interval, which again was not the usual procedure that was recommended for positive cases. These two aspects allowed us to identify unexpected patterns of viral load, immune responses and symptoms that would not normally be detected. We suggest that these findings may either be a result of a falsely positive PCR or alternatively of a transient infection. As both situations do not imply a relevant infection, our findings may have important implications for management strategies that are based on positive PCR tests.

In this cohort a relevant number of individuals had positive PCR test in swabs from the upper respiratory tract that did not result in a systemic response neither in symptoms nor in an immune response on both B cell and T cell level. In addition, viral replication was very low or possibly the result of mere colonization, as Ct levels were quite high and a negative follow-up PCR was detected at the next testing, which in most cases was performed the very next day. Of note, one individual (No. 4) developed an immune response (SARS-CoV-2-specific IgG, IgA, and CD4 T cells), although the viral load was rather low with the lowest Ct of all individuals (Ct 37.9), underscoring that not all samples with high Ct values above the analytical cut-off samples may have been false positives. Interestingly, another study among military personnel (*n* = 1453) also reported 14 cases of positive tests, of which 11 had transient episodes of PCR positivity followed by negative results in all subsequent tests [14]. As in our study, the authors also did not find any induction of antibodies, but the individuals were not further analyzed for induction of SARS-CoV-2 specific cellular immune response. Several explanations may seem plausible to explain a transiently positive PCR in asymptomatic individuals without induction of specific immune responses. A transiently positive PCR may be falsely positive. A false positive PCR test due to contamination during swabbing or poor quality of PCR testing cannot be excluded. However, the high number of individuals with this unusual pattern (at least 11 out of a maximum of 98 individuals) makes testing failure unlikely as the only explanation, especially in the context of high-quality dual target commercial PCRs that were performed in accredited laboratories and only considered positive if both targets reacted positive. Likewise, subsequent samples after the initial positive test result could have been falsely negative. One might even speculate that sampling could have been influenced to avoid exclusion of the player. However, this is considered unlikely, as PCR tests remained negative on several occasions. In addition, incorrect swabbing of an infected individual would not have prevented an immune response in 10 out of 11 individuals, where neither humoral nor cellular immune response was detectable. Another possible factor could be the development of mutations in the genome that interfere with primers or probes. However, these mutations would be more likely with a longer replication time. Here, the replication time was extremely short, so this phenomenon is considered rather unlikely. Finally, the immune tests used in our study may have failed to detect SARS-CoV-2 specific immune responses. However, we have used either commercial assays or in-house assays that have been extensively evaluated for characterization of the development of SARS-CoV-2 specific T cells after infection and COVID-19 vaccination [10,11,12,13,15,16].

Apart from mere technical issues, transient infections may represent another reason for transiently positive test results. In this situation, local components of the immune response may have limited further viral spread and subsequent induction of a systemic immune response, as has recently been suggested as a key mechanism why children are less affected by SARS-CoV-2 infection as compared to adults [17]. None of the individuals had any previous SARS-CoV-2 infection prior to the positive PCR reported here nor had they been vaccinated, as the vaccine was not available at the time of the study. Thus, the presumed local immunity could, on the one hand, be the result of innate immunity and, on the other hand, represent a local, not systemically detectable, cross-reactive immune response in the nasal mucosa resulting from a previous infection with one or more seasonal coronaviruses, mainly other betacoronaviruses such as HKU1 and OC43 [18,19]. A single positive PCR test in an asymptomatic individual should therefore be interpreted with caution, as it may reflect transient viral colonization limited to the upper respiratory tract [20] and does not necessarily represent a proof of a systemic infection in all cases. While the concept of resident and transient colonization of bacteria and their interplay with mucosal surfaces is well known both for commensal species and bacteria that cause clinically relevant diseases, the virome has more recently been characterized as consisting of mainly bacteriophages and viruses that infect other cellular microorganisms and/or human cells [21]. Approximately 10^8^ virus like particles per milliliter can be found in saliva, and the composition of viruses might be different between saliva and samples from the respiratory tract [22]. Only few studies focused on RNA viruses in the nasal or oral mucosa. Interestingly, however, a high number of different RNA viruses including coronaviruses were recently found in healthy children, showing that coronaviruses might be a part of the virome and thus might be present without causing symptoms [23].

The large number of individuals with frequent testing intervals and the close follow-up is considered a particular strength of our study. Nevertheless, we also identified some limitations. First, the observed phenomenon might even be more common in a real-world setting, as the league’s testing protocol had not encouraged retesting. Thus, the percentage of individuals with intermittently positive test results may have been underestimated. In addition, teams were not required to report retest results to the task force and generally did so only in cases of important players and negative PCRs in follow-up. Secondly, the real-world setting did not allow any retesting of swab samples by alternative tests as an independent confirmation. However, when developing the hygiene concept for the soccer teams, stringency of testing was ensured by the exclusive use of high-quality commercial dual target PCRs that were performed in accredited laboratories only. Thirdly, the period between the positive PCR and the collection of the blood sample ranged from 21 to 135 days. A minimum period of 21 days was considered important and sufficient for detection of humoral and cellular immune response. Although some parameters may decrease over time, specific IgG and T cell responses have been shown to persist for longer periods [24,25]. It is therefore considered unlikely, that we have missed detection of specific immune responses at later time points. Finally, the study was terminated in January 2021 because SARS-CoV-2 vaccines have since become available. This precluded further analyses, as vaccination leads to a preformed immune response and makes studies such as those conducted here impossible.

In summary, both falsely positive test results as well as transient infections may account for our observation of transiently positive PCRs in a substantial portion of individuals that were regularly tested. Further studies are needed to better distinguish between the two possibilities and to further characterize immunological mechanisms underlying transient infections. Interesting areas for future research may include characterization of local and systemic innate and adaptive cellular subsets associated with severe COVID-19, such as proliferative-exhausted CD8 T cells that may escape detection with functional assays [10,26], or NK cell subpopulations [27], or the quality and quantity of proinflammatory monocyte subpopulations [28], which may also play a role in local protection. In general, studies employing multi-omics may enable to correlate different factors and proteins with different clinical manifestations ranging from asymptomatic to severe disease [26,27,28]. It might be added that none of the positive tested individuals developed any symptoms including long COVID during a follow-up of at least 2 years. Although the number of individuals is rather small, this is in line with one of the hypotheses that (next to other factors) a dysregulated immune response might be a factor in the pathogenesis of long COVID and suggests that individuals without systemic immune response do not [29,30]. Whether this is because the virus was locally targeted in time or because no immune dysregulation was triggered is unknown, but in any case, worth further investigation.

## 5. Conclusions

In conclusion, our unique dataset acquired among a large cohort of professional football players and staff which underwent a strict regular PCR screening protocol shows that transiently positive PCRs with SARS-CoV-2 may frequently occur in asymptomatic individuals without systemic induction of a specific adaptive humoral or cellular immune response. This pattern found in around 10% of positive cases has so far likely been overlooked, since close monitoring and confirmatory retesting shortly after a positive PCR test is generally not recommended and thus rarely done. Further characterization of transient colonization regarding contagiousness is warranted, as measures such as treatment, isolation and contact tracing may not be justified in these cases. Our results indicate that under certain circumstances (low viral load and no symptoms), a positive PCR may not be interpreted as a proven infection that is normally followed by a range of consequences such as isolation, treatment, or the status of immunity as a recovered infection). This could also have implications in public health, since interpretation of a positive PCR is key in the management of future pandemics.

## Figures and Tables

**Figure 1 vaccines-11-00482-f001:**
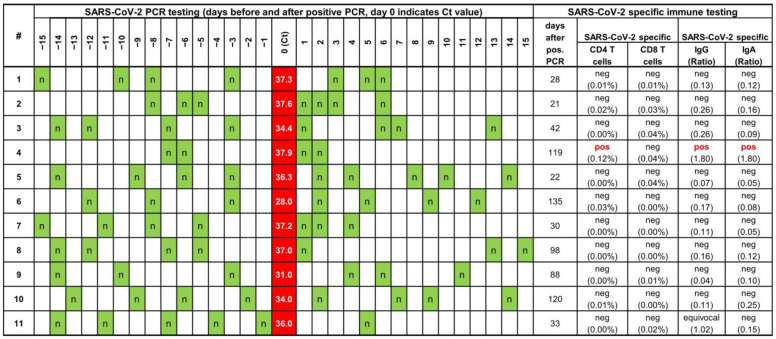
**Overview of SARS-CoV-2 PCR test results over time and SARS-CoV-2 specific humoral and cellular immune response.** Each line shows data from one individual. The chronological order of PCR testing before and after the positive test is shown with the positive PCR test result including Ct value labeled in red, and negative PCR results in green and indicated by an “*n*”. SARS-CoV-2 specific IgG and IgA were determined at a median of 42 (range 21–114) days after the positive PCR test. Individual #4 had negative serological results on screening 3 and 4 months before the positive PCR test, individual #11 had negative serological results on screening 6 months before the positive PCR. Ct, cycle threshold; IQR, interquartile range; PCR, polymerase chain reaction.

## Data Availability

All data of this study are listed in Figure 1.

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
