# Peer review of "Transient Positive SARS-CoV-2 PCR without Induction of Systemic Immune Responses"

_vaccines, 2023, doi:10.3390/vaccines11020482_

Round 1
Reviewer 1 Report
The manuscript ID: vaccines-2163148 “Transient infection with SARS-CoV-2 without induction of systemic immunity” by Gaertner and colleagues provides interesting insight in the SARS-CoV2 response of healthy individuals. It is understood that there is a limitation to access to samples due to the introduction of the vaccine. However, there are major weaknesses including an pitfalls in the study design (it reminds more of random sampling and analysis of what was available), underdeveloped introduction section, lack of information about the PCR assay, lack of confirmatory assays for virus presence such as an antigen assay, too limited analysis of the immune response, and an underdeveloped discussion. Some specific critique is listed in the following.
Abstract
Line 15: Specify how large the subset was that was retested after the initial positive PCR test.
Line 19: Indicate what was measured to determine systemic immunity. On which grounds did you determine that the single positive PCR test was not an artifact? It is suggested to delete this sentence if no rationale for this statement can be provided.
Introduction
The authors should expand on transient infections by viruses and compare to transient colonization by bacteria and address here local viral replication versus systemic spread.
The authors should also expand on local versus systemic immunity and what parameters are used to measure each.
The authors should explain why elite athletes were selected for this study.
The term systemic immunity should be replaced with systemic immune response- the subjects have not been intentionally challenged with SARS CoV2 to assess whether they have immunity.
Study protocol, Materials and Methods
Describe the study population more in detail with respect to age and sex.
The various subjects cannot be compared with respect to their immune response since their blood was analyzed at varying different times after the initial positive PCR test (between 21 and 135 days).
The authors should provide more information on the validity/specific/sensitivity of their test- was it an in-house test or commercial test? How were artifact excluded or identified among positive test results. Describe the targets and the actual sample processing, unless it is a kit from a commercial source.
Ig M should be also included in the assessment and the characterization of T cells is minimal and should be much more detailed.
The authors should have included subjects with repeatedly positive PCR and systemic immune response.
Discussion
The authors do not provide a strong rationale while false positive reactions are unlikely. Has an antigen test been conducted as a follow up? Or NGS?
The authors should discuss whether viral mutation within an individual may or may not be likely the underlying reasons for initial positive tests that are not confirmed in subsequent testing.
Author Response
Reviewer 1
The manuscript ID: vaccines-2163148 “Transient infection with SARS-CoV-2 without induction of systemic immunity” by Gaertner and colleagues provides interesting insight in the SARS-CoV2 response of healthy individuals. It is understood that there is a limitation to access to samples due to the introduction of the vaccine. However, there are major weaknesses including pitfalls in the study design (it reminds more of random sampling and analysis of what was available), underdeveloped introduction section, lack of information about the PCR assay, lack of confirmatory assays for virus presence such as an antigen assay, too limited analysis of the immune response, and an underdeveloped discussion. Some specific critique is listed in the following.
General response to reviewer 1:
We appreciate the summary and the comments provided by the reviewer. As a general comment to avoid misunderstanding on the study design, we would like to emphasize that our manuscript arose from a cohort of football players and staff that received mandatory regular PCR testing as part of the hygiene concept to ensure regular training and competitions, which constituted the setting for our study. Please note that our manuscript focuses on the closer follow-up of 11 individuals with a transient positive PCR, because this was considered unusual at the time. The reviewer is therefore correct in stating that analysis was restricted to these individuals only, whereas a closer follow-up of other individuals was not possible in the context of the study. As a result of this setting, many valid comments raised by the reviewers cannot be experimentally addressed further, but we have included them in the discussion as they provide useful areas for future research on this topic.
In response to the reviewer comments, we have therefore addressed general aspects and improved the introduction section, provided more information about the PCR assays including comments on the lack of confirmatory assays for virus presence, provided more details on analysis of the immune response, and the discussion was further expanded.
We also emphasize more openly that artifacts remain one potential reason for our observations on an individual basis, although we feel that the high number of individuals where a transiently positive PCR was observed provides some rationale for other explanations. We have now rephrased the text including the title to provide a more balanced view in interpreting the transiently positive PCRs as an artefact or a transient infection.
The specific responses to the reviewer´s comments are as follows:
Abstract: Line 15: Specify how large the subset was that was retested after the initial positive PCR test.
Unfortunately, we know the number of positive individuals (98) but not the exact number of positively tested players who were retested. The cohort was tested because the individuals were part of the "bubble" who were mandatorily tested in order to continue playing professional soccer, while retesting was neither recommended nor mandated. The club only retested in the event that the individuals (or their clubs) did not believe the result and the players wanted to participate in competitions. Thus, we do know that of the 98 individuals who tested positive, at least 11 opted to retest and were negative. As we cannot exclude that some others were also retested this percentage may be considered as a minimum. This had already been mentioned in the limitations section.
Abstract: Line 19: Indicate what was measured to determine systemic immunity. On which grounds did you determine that the single positive PCR test was not an artifact? It is suggested to delete this sentence if no rationale for this statement can be provided.
Thank you for this comment. We measured SARS-CoV2 specific antibodies (IgG and IgA) and SARS-CoV2 specific CD4 and CD8 positive T cells. Thus, we covered both the humoral and the cellular immune response. This information was now added to the abstract.
As stated in our general comment above, we followed the suggestion of the reviewer and deleted the sentence (line 19) while acknowledging that both artefacts as well as transient infections may account of our observations.
We agree with the reviewer, that the phenomenon of a single positive PCR without an immune response and symptoms can be interpreted in two ways. On the one hand, it may be an artifact (false positive test), on the other hand, the signal may be truely positive where another interpretation (e.g. local immune response) seems likely. With the data obtained, the two conditions cannot be distinguished with certainty. However, the tests were only performed in labs accredited according to the highest quality standard for medical laboratories (DIN ISO EN 15189, see also comment below). The labs used high quality commercial PCR assays, and a test result was only considered positive if at least two SARS-CoV-2 target genes were positive. The tested individuals were “super-healthy” as elite athletes. Moreover, the positive results we checked and reconfirmed together with the laboratories. TM and BG were members of the Coronavirus Task force of the DFL that was responsible for eligibility of the players, thus we had to check all positive results. Just to illustrate the situation: A positive PCR result in a professional football team is a big problem that has a major impact on the next competitions, the financial situation, and the media reports that covered every test result. This is why we checked the results with the laboratories for inaccuracies of the testing but could not find any.
Introduction: The authors should expand on transient infections by viruses and compare to transient colonization by bacteria and address here local viral replication versus systemic spread.
Thank you for pointing to this topic. To keep the introduction more stringent, we have now included this important topic in the discussion in more detail.
It now reads as follows:
“…While the concept of resident and transient colonization of bacteria and their interplay with mucosal surfaces is well known both for commensal species and bacteria that cause clinically relevant diseases, the virome has more recently been characterized as consisting of mainly bacteriophages and viruses that infect other cellular microorganisms and/or human cells (21). Approximately 108 virus like particles per millilitre can be found in saliva, and the composition of viruses might be different between saliva and samples from the respiratory tract (22). Only few studies focused on RNA viruses in the nasal or oral mucosa. Interestingly, however, a high number of different RNA viruses including coronaviruses were recently found in healthy children, showing that coronaviruses might be a part of the virome and thus might be present without causing symptoms (23)….”
The authors should also expand on local versus systemic immunity and what parameters are used to measure each.
We followed the suggestions of the reviewer and have now included a section of the role of innate components of the immune system, that controls local infections. Before induction of specific immunity. This now reads as follows:
The local and the systemic immunity is a result of complex mechanisms involving the innate and the adaptive immune system and consists of humoral as well as cellular factors. Whereas systemic responses are usually measured in blood, local reactions are much more difficult to determine. Local immunity is a barrier that blocks further spread of infection which may not necessarily result in induction of a systemic immune response. The innate local immunity against RNA viruses is usually stimulated by the presence of viral RNA which is recognized by various pattern recognition receptors. This is followed by a cascade of events including secretion of cytokines or chemokines followed by further attraction of other cells including neutrophils, monocytes and natural killer (NK) cells. Only if innate immunity itself is not sufficient in blocking the infection, the virus will establish infection in the cell, and adaptive immune responses are induced. The adaptive immune response is dominated by antigen-specific antibodies and antigen-specific T cells, which are detectable in circulation but also play a role at the site of infection (7).
Introduction: The authors should explain why elite athletes were selected for this study.
We observed the phenomenon in the cohort of elite athletes tested as part of a particular hygiene concept as mentioned above. Although this could have been tested in any other cohort, the setting of elite athletes had some unique advantages that allowed us to identify the phenomenon of transient positive PCRs and perform the follow-up studies: Without close PCR testing (2x/week), the duration of PCR positivity could not have been determined. Without re-testing of the initially positive individuals (which is usually not done in other settings), the phenomenon could not be detected either.
This has been explained in more detail in the “Methods” section.
The term systemic immunity should be replaced with systemic immune response- the subjects have not been intentionally challenged with SARS CoV2 to assess whether they have immunity.
We now have modified the terminology accordingly.
Study protocol, Materials and Methods: Describe the study population more in detail with respect to age and sex.
As stated above please note, that our manuscript mainly deals with the 11 individuals that showed a transiently positive PCR among a larger cohort of approximately 1700 elite football players and staff, that had been described in more detail in a previous study (Mack et al.). We added some general information to the “Methods” section. This now reads as follows:
“…The study participants with a transiently positive PCR originated from a cohort of elite football players and staff, where a special hygiene program was established in May 2020 during the first lockdown in Germany to facilitate the restart of the German Bundesliga and Bundesliga 2 with matches behind closed doors (8, 9). This cohort comprised approximately 1,700 players and staff (mostly males (>95%), median age of approximately 30 years as described before (8)) with direct contact to players (e.g., coaches, physiotherapists) of the 36 professional soccer teams in Germany.
Study protocol, Materials and Methods: The various subjects cannot be compared with respect to their immune response since their blood was analyzed at varying different times after the initial positive PCR test (between 21 and 135 days).
We thank the reviewer for this comment. The following sentence was now added in the methods section: “The interval between positive PCR and blood sample had to be at least 21 days to give the immune response sufficient time to develop.”
Moreover, the following sentence was added in the discussion in the limitations section: “…Second, the period between the positive PCR and the collection of the blood sample ranged from 21 to 135 days. A minimum period of 21 days was considered important and sufficient for detection of humoral and cellular immune response. Although some parameters may decrease over time, specific IgG and T-cell responses have been shown to persist for longer periods.” It is therefore considered unlikely, that we have missed detection of specific immune responses at later time points.
The authors should provide more information on the validity/specific/sensitivity of their test- was it an in-house test or commercial test? How were artifacts excluded or identified among positive test results. Describe the targets and the actual sample processing, unless it is a kit from a commercial source.
The antibody assays and the PCRs were commercial test as now indicated in the M&M section. The PCR tests are now specified in more detail. The T cells assays were developed in our group as an in-house test but extensively evaluated and used in a large number of studies on individuals after SARS-CoV-2 infection and vaccination. Here are some of the publications:
- Schub D, Klemis V, Schneitler S, Mihm J, Lepper PM, Wilkens H, Bals R, Eichler H, Gärtner BC, Becker SL, Sester U, Sester M, Schmidt T. High levels of SARS-CoV-2-specific T cells with restricted functionality in severe courses of COVID-19. JCI Insight. 2020 Oct 15;5(20):e142167.
- Klemis V, Schmidt T, Schub D, Mihm J, Marx S, Abu-Omar A, Ziegler L, Hielscher F, Guckelmus C, Urschel R, Wagenpfeil S, Schneitler S, Becker SL, Gärtner BC, Sester U, Sester M. Comparative immunogenicity and reactogenicity of heterologous ChAdOx1-nCoV-19-priming and BNT162b2 or mRNA-1273-boosting with homologous COVID-19 vaccine regimens. Nat Commun. 2022 Aug 11;13(1):4710.
- Schmidt T, Klemis V, Schub D, Mihm J, Hielscher F, Marx S, Abu-Omar A, Ziegler L, Guckelmus C, Urschel R, Schneitler S, Becker SL, Gärtner BC, Sester U, Sester M. Nat Med. 2021 Sep;27(9):1530-1535.
- Hielscher F, Schmidt T, Klemis V, Wilhelm A, Marx S, Abu-Omar A, Ziegler L, Guckelmus C, Urschel R, Sester U, Widera M, Sester M. NVX-CoV2373-induced cellular and humoral immunity towards parental SARS-CoV-2 and VOCs compared to BNT162b2 and mRNA-1273-regimens. J Clin Virol. 2022 Dec;157:105321.
IgM should be also included in the assessment and the characterization of T cells is minimal and should be much more detailed.
We thank the reviewer for this suggestion. We agree that additional testing of IgM could add some more information, however we have chosen IgA, as it has been shown to persist longer than IgM (i.e. Gaebler et al. Nature 2021). Since the goal of the follow-up test was to find an immune response, we feel that results from IgA can replace IgM here.
Apart from the methodological details, we have now described the T cell tests which is based on intracellular cytokine staining using flow cytometry in more detail and also added some more references for the T cell test.
The authors should have included subjects with repeatedly positive PCR and systemic immune response.
While this was not possible in the setting of the current study, we and others have performed numerous studies characterizing the development of antibody and T-cell responses after both infection and vaccination. Apart from vaccine-induced immune-responses, these studies have shown that immunocompetent individuals with positive PCR and/or antigen tests consistently develop SARS-CoV-2 specific antibodies and T-cells. These studies on the induction of specific immunity based on the same assays are now included in the discussion.
Discussion: The authors do not provide a strong rationale while false positive reactions are unlikely. Has an antigen test been conducted as a follow up? Or NGS?
We have now further expanded the discussion on potential reasons for our observations of transiently positive PCR results. We discuss both the possibility of an artefact as well as a true transiently positive results and disclose more openly that both possibilities cannot be distinguished on an individual basis. As mentioned earlier, based on the high number of subjects in which the phenomenon occurred, combined with the high quality of the tests, artifacts seem unlikely as the only reason for our observations.
An antigen test or NGS was unfortunately not performed on these samples. When developing the hygiene concept for the soccer teams, PCR was the first assay available and chosen as the most sensitive test method for both initial testing and repeat testing on follow-up. Although other assay principles would have been valuable as independent confirmation, this was unfortunately not performed at that time, because antigen tests were either not yet available or considered less sensitive as a confirmative test. This lack of further testing was now included in the discussion as a limitation.
Discussion: The authors should discuss whether viral mutation within an individual may or may not be likely the underlying reasons for initial positive tests that are not confirmed in subsequent testing.
Thank you for this interesting point, which was now included in the discussion as follows: “Another possible factor could be the development of mutations in the genome that interfere with primers or probes. However, these mutations would be more likely with a longer replication time. Here, the replication time was extremely short, so this phenomenon is rather unlikely.”

Reviewer 2 Report
Gärtner et al analyzed 1700 individuals who were regularly tested by PCR twice weekly, where 98 individuals had a positive PCR and they identified, among subsets of these who were retested shortly after the initial positive result, 11 subjects were who only had a transient single positive PCR of low viral load. They found that only one out of 11 individuals developed SARS-CoV-2 specific cellular and humoral immunity after the positive PCR, whereas a specific immunity was undetectable in all other individuals. Thus, the authors conclude that a single positive PCR might indicate that transient colonization of the upper respiratory tract with SARS-CoV-2 may occur without systemic induction of specific adaptive immunity. Overall, this hypothesis is possible but the evidence provided by the authors cannot fully support their conclusion. Meanwhile, the study in its current format is shallow with only a few simple assays. Thus, a major revision addressing all of the following points will be needed.
1. One of the key evidence the authors use to define if a patient is positive is based on PCR test of swab samples. However, the cut-off for PCR test matters. Could the authors try different and maybe more stringent cut-offs to define positive patients and see if only patient no.4 will be positive and others will mostly not be positive anymore? If that’s the case, then the previously defined 11 positive patients may not be really positive. It could be that they are just having a noisy swab sample. Similarly, if there is an independent way to tell if the patient is positive, that needs to be conducted on the 11 patients to confirm that they are truly positive. This is the key evidence that the major conclusion was based on, so it requires more stringent evaluations.
2. The molecular assays the author performed are limited. It will be much more interesting if the authors could conduct some more multi-omic analysis on the patients’ PBMC and/or serum to deeply understand for example what are the many immunological differences between patient No.4 vs other 10 individuals that did note develop systematic immunity. Such multi-omic difference can also help infer some of the underpinning mechanisms of how to generate a good protection that restrict the virus to only locally. In fact, such deep multiomic molecular characterization methodologies has already been conducted to study COVID-19 severity (e.g. PMID:33171100, 34592166, 33765435), mortality (e.g. PMID: 34489601) and long COVID (e.g. PMID: 35216672, 35258337). If it is hard to perform such molecular characterizations, the authors should at least describe such as a future direction in the discussion section of the paper, with those relevant literature cited and discussed.
3. Do any of these 11 patients develop long covid? If so, it will be interesting to evaluate those patients’ immune profiles similar to some of the recent long COVID work (e.g. PMID: 35216672, 35258337). If not possible to perform, this should be discussed at least as an important future direction within the context of the appropriate literature.
4. The authors suggested that “Local components of the immune response may have limited further viral spread and subsequent induction of a systemic immune response”. This is an interesting perspective. It will be interesting to investigate what are the specific immune cells that contributed or “anti-contribute” to such protections. There are many recent literatures that give specific examples of a few novel immune phenotypes that have been identified in COVID-19. Those specific novel immune phenotypes could be investigated or at least specifically discussed within the context of this study. For example, there have been recent reports of proliferative-exhausted CD8 T cells (PMID: 33171100) which seem to be a novel phenotype that is strongly associated with COVID-19 severity and potentially in long covid. Discussion of their role within the context of this local protection theory as a future direction of investigation could improve the depth of the current manuscript. Another example is that there has been recent literatures (e.g. PMID: 34489601) reporting that monocytes bifurcated into two subpopulations pro-inflammatory and anti-inflammatory subpopulations and they found the quality and quantity of proinflammatory subsets are increased with covid-19 severity. Will any of these monocytes be contributing to local protection? This can be discussed to further elevate the depth of this paper. Similarly, there has already been recent literature on the novel “adaptive NK phenotype” (PMID: 36066491) and on NK cells with unique early IFNa signatures (PMID: 34592166). These two NK phenotypes are both closely associated with covid-19 severity. It will be one of the important future directions to further evaluate these NK phenotypes in their transient positive cohort to study these novel phenotypes’ association with protections.
5. Biomarkers to tell which patients with a transient positive test will later develop systematic immunity. It will be interesting to evaluate some of the blood biomarker differences between the patient No.4 and the rest of the 11 patients. These can be useful for pre-classifying these two different types of patients. Proteomics and metabolomics can be great methods for identifying such biomarkers, for example PMID: 34489601 has used such methods to identify death-related biomarkers. This should be investigated if possible. If not, at least discussed as an important future direction within the context of those literature.
6. Accessibility of de-identified data. The data collected from this paper is informative to the field of covid research. It will be important for the authors to provide their de-identified data (e.g. T cell assay, antibody assay, PCR test etc. ) as supplementary tables. These data is helpful for the reviewers to evaluate their work, and this is also important later for the public to utilize these data to investigate their own hypothesis of interest since some of their more complex hypotheses could be beyond the current manuscript’s scope.
Author Response
Reviewer 2
Gärtner et al analyzed 1700 individuals who were regularly tested by PCR twice weekly, where 98 individuals had a positive PCR and they identified, among subsets of these who were retested shortly after the initial positive result, 11 subjects were who only had a transient single positive PCR of low viral load. They found that only one out of 11 individuals developed SARS-CoV-2 specific cellular and humoral immunity after the positive PCR, whereas a specific immunity was undetectable in all other individuals. Thus, the authors conclude that a single positive PCR might indicate that transient colonization of the upper respiratory tract with SARS-CoV-2 may occur without systemic induction of specific adaptive immunity. Overall, this hypothesis is possible but the evidence provided by the authors cannot fully support their conclusion. Meanwhile, the study in its current format is shallow with only a few simple assays. Thus, a major revision addressing all of the following points will be needed.
General response to reviewer 2:
We appreciate the summary and the comments provided by the reviewer. As a general comment to avoid misunderstanding on the study design, we would like to emphasize that our manuscript arose from a cohort of soccer players and staff that received mandatory regular PCR testing as part of the hygiene concept to ensure regular training and competitions, which constituted the setting for our study. Please note that our manuscript focuses on the closer follow-up of 11 individuals with a transient positive PCR, because this was considered unusual at the time. The reviewer is therefore correct in stating that analysis was restricted to these individuals only, whereas a closer follow-up of other individuals was not possible in the context of the study. As a result of this setting, many valid comments raised by the reviewers cannot be experimentally addressed further, but we have included them in the discussion as they provide useful areas for future research on this topic (see also comments to reviewer 1).
We also emphasize more openly that artifacts remain one potential reason for our observations on an individual basis, although we feel that the high number of individuals where a transiently positive PCR was observed provides some rationale for other explanations. We have now rephrased the text including the title to provide a more balanced view in interpreting the transiently positive PCRs as an artefact or a transient infection.
- One of the key evidence the authors use to define if a patient is positive is based on PCR test of swab samples. However, the cut-off for PCR test matters. Could the authors try different and maybe more stringent cut-offs to define positive patients and see if only patient no.4 will be positive and others will mostly not be positive anymore? If that’s the case, then the previously defined 11 positive patients may not be really positive. It could be that they are just having a noisy swab sample. Similarly, if there is an independent way to tell if the patient is positive, that needs to be conducted on the 11 patients to confirm that they are truly positive. This is the key evidence that the major conclusion was based on, so it requires more stringent evaluations.
We thank the reviewer for this valid comment. Unfortunately, the real world setting did not allow any retesting of swab samples by alternative tests as an independent confirmation, which we have acknowledged in the limitations section.
Please note, however, when developing the hygiene concept for the soccer teams, stringency of testing was ensured by the exclusive use of high quality commercial dual target PCRs that were performed in accredited laboratories (DIN ISO EN 15189) only. The PCR assays that were used have now been specified in the methods section. Test results were interpreted based on the analytical cut-offs determined by the manufacturers. The analytical cut-off should describe a value that differentiates best between signal and no signal. It is balanced between sensitivity and specificity and defines the limit of detection that is stated by the manufacturer. Since this is done by the manufacturer and is part of the approval of assays, it is difficult to use other cut-offs and to be sure that no RNA is present although the test revealed a clear signal. This is why we choose to work with the analytical cut-off.
Analytical accuracy of the dual target PCRs for a true detection of SARS-CoV-2 RNA is considered high, as both targets were required to be positive, which should reduce the number of subjects with “noisy” samples. While the clinical consequences of PCR tests with a high Ct value may be clear and may be considered not meaningful, a rationale to interpret a high Ct value as “noisy” rather than acknowledging the presence of low level RNA, is considered scientifically difficult.
- The molecular assays the author performed are limited. It will be much more interesting if the authors could conduct some more multi-omic analysis on the patients’ PBMC and/or serum to deeply understand for example what are the many immunological differences between patient No.4 vs other 10 individuals that did note develop systematic immunity. Such multi-omic difference can also help infer some of the underpinning mechanisms of how to generate a good protection that restrict the virus to only locally. In fact, such deep multiomic molecular characterization methodologies has already been conducted to study COVID-19 severity (e.g. PMID:33171100 (1), 34592166 (2), 33765435(3), mortality (e.g. PMID: 34489601(4) and long COVID (e.g. PMID: 35216672 (5),, 35258337(6)). If it is hard to perform such molecular characterizations, the authors should at least describe such as a future direction in the discussion section of the paper, with those relevant literature cited and discussed.
Thank you very much for these suggestions. We acknowledge that further analyses in this direction were not possible in the setting of the study, but we agree that these represent interesting suggestions to provide a deeper understanding on the observed phenomenon. This was now included in the discussion.
- Do any of these 11 patients develop long covid? If so, it will be interesting to evaluate those patients’ immune profiles similar to some of the recent long COVID work (e.g. PMID: 35216672m (5), 35258337(6). If not possible to perform, this should be discussed at least as an important future direction within the context of the appropriate literature.
No, all individuals were completely asymptomatic and did not develop any signs of COVID. Based on their continued professional activity as elite athletes (nine), coaches (two) or physiotherapists (one) we do not have any evidence for occurrence of long COVID.
- The authors suggested that “Local components of the immune response may have limited further viral spread and subsequent induction of a systemic immune response”. This is an interesting perspective. It will be interesting to investigate what are the specific immune cells that contributed or “anti-contribute” to such protections. There are many recent literatures that give specific examples of a few novel immune phenotypes that have been identified in COVID-19. Those specific novel immune phenotypes could be investigated or at least specifically discussed within the context of this study. For example, there have been recent reports of proliferative-exhausted CD8 T cells (PMID: 33171100(1)) which seem to be a novel phenotype that is strongly associated with COVID-19 severity and potentially in long covid. Discussion of their role within the context of this local protection theory as a future direction of investigation could improve the depth of the current manuscript. Another example is that there has been recent literatures (e.g. PMID: 34489601 (2)) reporting that monocytes bifurcated into two subpopulations pro-inflammatory and anti-inflammatory subpopulations and they found the quality and quantity of proinflammatory subsets are increased with covid-19 severity. Will any of these monocytes be contributing to local protection? This can be discussed to further elevate the depth of this paper. Similarly, there has already been recent literature on the novel “adaptive NK phenotype” (PMID: 36066491(7)) and on NK cells with unique early IFNa signatures (PMID: 34592166(3)). These two NK phenotypes are both closely associated with covid-19 severity. It will be one of the important future directions to further evaluate these NK phenotypes in their transient positive cohort to study these novel phenotypes’ association with protections.
We thank the reviewer for providing an interesting set of additional parameters that may be observed after contact with SARS-CoV-2. Apart from the most prominent parts of adaptive immune responses, i.e. antibodies and CD4 and CD8 T cells, those may be interesting for future study as potentially interesting independent indicators for either local immunity or systemic immune responses typically found in individuals with COVID-19. Some interesting subsets including references have now been added to the discussion.
- Biomarkers to tell which patients with a transient positive test will later develop systematic immunity. It will be interesting to evaluate some of the blood biomarker differences between the patient No.4 and the rest of the 11 patients. These can be useful for pre-classifying these two different types of patients. Proteomics and metabolomics can be great methods for identifying such biomarkers, for example PMID: 34489601 (4) has used such methods to identify death-related biomarkers. This should be investigated if possible. If not, at least discussed as an important future direction within the context of those literature.
As stated above, we are unfortunately unable to perform some more detailed analyses. However, we thank the reviewer for his/her suggestions on how to enrich the discussion with these interesting areas for future research. This was now also added to the discussion.
- Accessibility of de-identified data. The data collected from this paper is informative to the field of covid research. It will be important for the authors to provide their de-identified data (e.g. T cell assay, antibody assay, PCR test etc.) as supplementary tables. These data is helpful for the reviewers to evaluate their work, and this is also important later for the public to utilize these data to investigate their own hypothesis of interest since some of their more complex hypotheses could be beyond the current manuscript’s scope.
Please note that most of the data were already included in Table 1 (test results i.e. samples/cut-off ratio for antibodies, or % of positive cells in T cell assays, Ct values) and their interpretation as positive or negative. As the whole manuscript focused on the analysis of these 11 individuals shown in table 1, we do not see further room for providing more detailed information but would be happy to do so if deemed helpful.

Round 2
Reviewer 1 Report
This reviewer appreciates the thoughtful response of the authors and the corrections made. The revised manuscript has addressed most of the concerns raised. There is one minor request: Please add the exact number of study participants in the cohort in the methods section (line 75). Each subject should be accounted for when performing human subjects (or animal) studies.
Reviewer 2 Report
This revised manuscript is improved after accounting for some of the suggestions from this reviewer. However, a few of the important points that this reviewer suggested in the initial review report was not adequately addressed. As a result, the current manuscript is still not at a level that is suitable for publication. Thus, this reviewer will suggest another round of revision to address some of the remaining points adequately.
1. This reviewer can understand why the authors first worked with the manufacturer's suggested analytical cut-off. However, this reviewer still thinks it will be important to emphasize the point that patient4, which is the only one that had SARS-CoV2 specific T cell and antibody responses, in fact, showed the highest CT values (lowest viral load). This may reflect the possibility that the authors’ argument is indeed possible.
2. The fact that none of these individuals develop long covid is very interesting and definitely worth including into the discussion section of the paper and discussed within the context of some of the recent literature (e.g. PMID: 35216672, 35258337). These may indicate that if there is no systematic immune response post PCR positivity, then the long covid chances are lower. Whether this is because the virus has been timely controlled locally (so the virus was not causing certain long covid symptoms in some other organs) or because no dys-wiring of the immunity has been triggered (so dysregulation of the immune systems is not causing certain long covid symptoms), is not known but definitely worth further efforts of investigation.
3. Although most of the data is provided in Table 1, it will be worthwhile to provide the CT values of all 11 patients across time points to evaluate if different cut-offs of CT values may show some different patterns on patient 4 versus the other 10 patients.
